

# Detectability of forced trends in stratospheric ozone

Louis Rivoire[1,2,3], Marianna Linz[1,2], Jessica L. Neu[4], Pu Lin[5], and Michelle L. Santee[4]

[1]Department of Earth and Planetary Sciences, Harvard University, Cambridge, Massachusetts, USA
[2]School of Engineering and Applied Sciences, Harvard University, Cambridge, Massachusetts, USA
[3]Department of Earth, Atmospheric, and Planetary Sciences, Massachusetts Institute of Technology, Cambridge, Massachusetts, USA
[4]Jet Propulsion Laboratory, California Institute of Technology, Pasadena, California, USA
[5]Program in Atmospheric and Oceanic Sciences, Princeton University, Princeton, New Jersey, USA

**Correspondence:** Louis Rivoire (lrivoire@mit.edu)

**Abstract.** The continued monitoring of the ozone layer and its long-term evolution leans on comparative studies of merged satellite records. Such records present unique challenges due to differences in sampling, coverage, and retrieval algorithms between observing platforms, leading to discrepancies in trend calculations. Here we examine the effects of optimal estimation retrieval algorithms on vertically resolved ozone trends, using one merged record as an example. We find errors as large as

1 % per decade and displacements in trend profile features of as much as 6 km altitude due to the vertical redistribution of information by averaging kernels. Furthermore, we show that averaging kernels tend to increase the length of record needed to determine whether vertically resolved trend estimates are distinguishable from natural variability with good statistical confidence. We conclude that trend uncertainties may be underestimated, in part because averaging kernels misrepresent decadal to multi-decadal internal variability, and in part because the removal of known modes of variability from the observed record can

yield residual errors. The study provides a framework to reconcile differences between observing platforms, and highlights the need for caution when using merged satellite records to quantify trends and their uncertainties.

## 1  Introduction

Since the discovery of the Antarctic ozone hole (Farman et al., 1985) and the advent of the World Meteorological Organization (WMO)/United Nations Environment Programme (UNEP) Ozone Assessment reports in 1989, there have been a number

of activities dedicated to monitoring the state of the ozone layer and attributing long-term changes in ozone distribution. Notwithstanding declining levels of ozone-depleting substances (ODSs) (WMO, 2022), the search for evidence of the continued recovery of the global ozone layer is ongoing. A number of chemistry-climate feedbacks complicate this search. The greenhouse-driven cooling of the stratosphere is thought to aid the recovery in the mid-to-upper stratosphere by slowing down reactions that deplete ozone (Fels et al., 1980). In contrast, the acceleration of the Brewer-Dobson circulation widely predicted

by climate-chemistry models (e.g., Eyring et al., 2013a; Meul et al., 2014; Abalos et al., 2021) is expected to redistribute lower stratospheric ozone from the tropics to the midlatitudes (e.g., Rind et al., 1990; Mahfouf et al., 1994; Shepherd, 2008) and to decrease lower stratospheric ozone abundances (Shepherd, 2008; Li et al., 2009; Waugh et al., 2009; SPARC, 2010; Oman et al., 2010; Plummer et al., 2010; Meul et al., 2014; Banerjee et al., 2016; Chiodo et al., 2018; Dietmüller et al., 2021), with



large impacts on surface warming (Nowack et al., 2015) and ultraviolet radiation (Hegglin and Shepherd, 2009). Similarly, the
expansion of the troposphere was also found to decrease stratospheric ozone by eroding the ozone layer from below (Match and
Gerber, 2022). Because of these feedbacks, projected rates of recovery from climate-chemistry models are sensitive to model
formulation (Dietmüller et al., 2014) and greenhouse gas emission scenarios (Revell et al., 2012). Lastly, emerging influences
are adding new complexity to our understanding of the composition of the stratosphere, for instance, large wildfires and asso-
ciated aerosols (Santee et al., 2022; Solomon et al., 2023), newly detected ODS emissions (e.g., Montzka et al., 2018; Rigby
et al., 2019; Chipperfield et al., 2020), new estimates for ODS lifetimes (Lickley et al., 2021), increasingly large dynamical
variability (Diallo et al., 2018, 2019), and large volcanic eruptions (WMO, 2022; Wang et al., 2023; Evan et al., 2023; Santee
et al., 2023; Wilmouth et al., 2023; Manney et al., 2023).

     In light of such complexity, the analysis of observed trends continues to present challenges. The last WMO report (WMO,
2022) found a small but positive trend ($0.3 \pm 0.2$ %/decade) in the near-global (60°S-60°N) total column ozone. Vertically
resolved trends reveal regional patterns hinting at competing influences on ozone abundances. The analysis of merged satel-
lite products (e.g., Sofieva et al., 2017; Steinbrecht et al., 2017; Ball et al., 2017; Bourassa et al., 2018; WMO, 2018;
Petropavlovskikh et al., 2019) found large positive post-2000 trends across the upper stratosphere (above 5 hPa), in agree-
ment with the climate-chemistry model initiative (CCMI) simulations (Eyring et al., 2013b; Godin-Beekmann et al., 2022).
However, agreement is lacking in the lower stratosphere ($50 - 10$ hPa): negative though uncertain trends are found in several
satellite records, and though CCMI trends agree in sign below 30 hPa (Petropavlovskikh et al., 2019), the range across models
and even across ensemble members of a single model is large (Stone et al., 2018). In the lowermost stratosphere ($50 - 100$
hPa), models and observations are in disagreement and trends remain uncertain. In addition, reanalysis products present vary-
ing degrees of realism in ozone trends (Davis et al., 2017). Whether models are flawed in their representation of stratospheric
ozone or whether the statistical confidence placed in trend estimates is sufficient to address disagreements between models and
observations, the need to improve our ability to distinguish trends from internal variability is clear.

     In this paper, we turn to the challenges inherent to merged records: differences in retrieval methods, coverage, and resolution
across platforms, varying merging techniques, etc. Of particular interest are the nontrivial effects of error propagation in
retrieval algorithms. Such effects are generally known to reduce the statistical confidence placed in trend estimates from merged
records (Tummon et al., 2015; Hubert et al., 2016; Steinbrecht et al., 2017; Ball et al., 2017). This is especially concerning given
that trends originally assigned high statistical confidence can still suddenly change magnitude or even sign after the addition
of just a few years of record (Chipperfield et al., 2018; Ball et al., 2019). Past literature provides extensive discussion about
quantifying and reducing uncertainty in ozone trends (e.g., Stolarski and Frith, 2006; Harris et al., 2015; Petropavlovskikh
et al., 2019), but methods have not yet been developed to systematically account for the limitations of satellite records. Using
the example of the SBUV merged record (McPeters et al., 2013), we utilize a novel method (Rivoire et al., 2024) to determine
whether current trend estimates are distinguishable from natural variability, particularly when accounting for the effects of
errors attributable to the SBUV averaging kernels.



## 2 Model output and observational data

In this section we introduce the data sets used in this study, namely:

- A pre-industrial chemistry-climate control simulation used as reference for internal variability;

- Chemistry-climate simulations used to estimate the time of emergence of future long-term ozone trends;

- The SBUV merged record, used as reference for the effects of satellite kernels;

- Other merged satellite records, used to compare against SBUV.

### 2.1 Simulations used as reference for internal variability

In order to determine whether a trend derived from the observational record reflects natural variability or arises because of

anthropogenic forcings, we need to compare said trend to a reference distribution of naturally occurring trends in a pre-industrial setting. Since the satellite era only covers the past 40-50 years, satellite records cannot be used to determine such a reference distribution. We therefore turn to a simulation of the pre-industrial era from a chemistry-climate model.

Two runs from the Geophysical Fluid Dynamics Laboratory's Earth System Model version 4.1 (ESM4.1, Dunne et al., 2020) are used as reference: 1) a 500-year pre-industrial run ("piControl" in CMIP6 nomenclature), and 2) a historical run from 1850

to the end of 2014 ("historical"). The historical run provides a benchmark for the model's performance given forcings and boundary conditions matching the historical record, including aerosol optical depths, 43 short-lived and long-lived greenhouse gases (GHGs), land use, solar forcing, sea surface temperatures, and sea ice concentrations. The pre-industrial run excludes such time-dependent forcings and is instead run with a prescribed global annual mean atmospheric $CO_2$ concentration equal to that of 1850 (Eyring et al., 2016) and with a time invariant volcanic aerosol forcing equivalent to the historical average (1850-

2014). Solar irradiance is also maintained constant following the specifications for the CMIP6 pre-industrial run (documented at http://goo.gl/r8up31). The Quasi-biennal Oscillation (QBO) in the model is internally generated.

The ESM4.1 model is the product of development efforts to capture coupled ocean-atmosphere and land-atmosphere inter-actions, biogeochemical cycles and ecosystem physics, sea ice, aerosol processes, and—importantly for this study—interactive ozone chemistry. The ozone chemistry includes an improved representation of ozone precursors (methane, carbon monoxide,

nitrogen oxides, and volatile organic compounds) and accounts for heterogeneous reactions that occur on the surface of aerosols (Austin et al., 2013). ESM4.1 features 49 vertical levels with a model top at 1 Pa (∼80 km) to better capture stratospheric chemistry and dynamics compared to the previous model generation (Horowitz et al., 2020). Coupled model simulations rarely include fully interactive chemistry in their extended control runs, making the ESM4.1 control run a unique asset for our study.

In order to ensure that the simulation provides suitable reference trend distributions, we assess the realism of the model's

representation of stratospheric ozone variability. Simulated and observed variability are portrayed by power density spectra in Figure 1. The spectra are estimated using the multitaper method (Thomson, 1982) that is suitable for short records. On interannual to multidecadal scales, it is inherently difficult to test whether the ozone variability produced by the model under pre-industrial conditions is realistic, since we have no historical record that spans the pre-industrial era. However, we can test





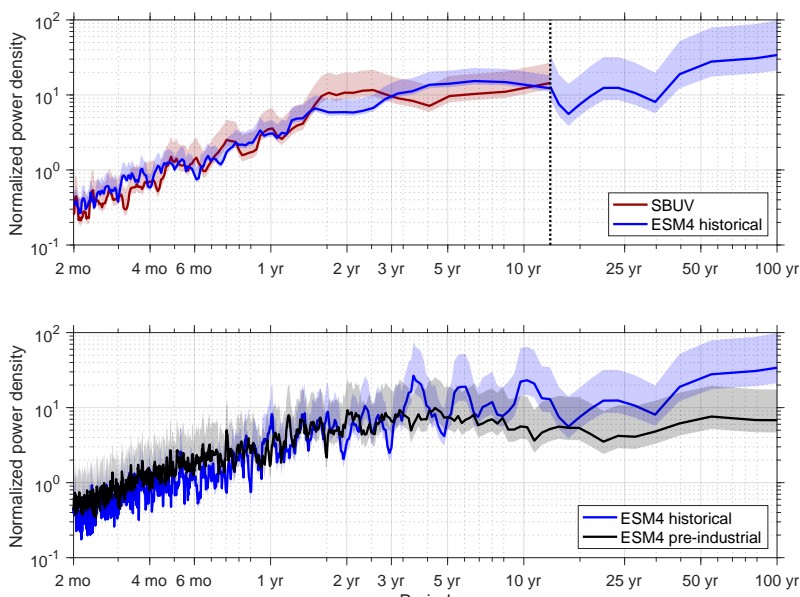

**Figure 1.** Comparison between observed and simulated normalized variability in the de-seasonalized zonal mean ozone for the 10-16 hPa layer in the Northern midlatitudes (37.5°N-57.5°N), pictured by power spectral densities (PSD) estimated with the multitaper method. Shaded areas indicate chi-squared 95 % confidence intervals. To ensure comparability between spectra, the ESM4.1 historical run (1850-2014) is sub-sampled like SBUV (1970-2020) for periods shorter than the longest continuous period given by the SBUV record (~13 years, dashed line) and the resulting power spectra are subsequently averaged. Similarly, the ESM4.1 pre-industrial run (500 years) is sub-sampled like the historical run in the bottom panel. Further, spectral density is normalized by the area under each power spectrum to remove changes in total power.

whether the simulated ozone variability is on par with observations when the model is provided with historical radiative and

chemical forcings that correspond to greenhouse gas and ODS emissions. If the model can capture the historical variability, we can at least say that it is skilled in capturing processes that are relevant to our current understanding of the atmosphere. One caveat in this reasoning is the fact that model improvements are tested against the historical record, which contains anthropogenic forcings, i.e., it is possible that some model improvements target modes of variability that are specific to forcings rather than to natural climate variability. That being said, the historical model simulation exhibits improvements in the representation

of total column ozone interannual variability and trends over previous versions of the model (Horowitz et al., 2020).

The pre-industrial simulation produces less ozone variability on decadal to secular periods than the historical run. This behavior is expected since the pre-industrial run excludes forcings that affect ozone on these time scales (ODSs, GHGs, volcanic eruptions). This result means that trend uncertainties estimated from these simulations are likely to be optimistic.



## 2.2 Simulations used to estimate trends

We use the average ozone trends from 2000-2020 from the same data set as in Godin-Beekmann et al. (2022): the multi-model mean (16 models) from the CCMI's 1960-2100 period (REF-C2, Eyring et al., 2013b). The purpose of the REF-C2 experiments is to capture future climate trends, and as such they follow the WMO (2011) A1 scenario for ODSs and the RCP6.0 for other greenhouse gas, ozone precursor, and aerosol precursor emissions. Depending on model capabilities, ocean conditions and forcings for the 11-year solar cycle and the QBO are either simulated separately or internally generated. Though the recommendation was not to use volcanic forcings, some models do include them (Godin-Beekmann et al., 2022). More information about the forcings can be found in Eyring et al. (2013a); Hegglin et al. (2016); Morgenstern et al. (2017).

## 2.3 SBUV satellite record

The merged SBUV dataset (MOD, version 2 release 1, https://acd-ext.gsfc.nasa.gov/Data_services/merged/) combines retrievals from back-scattered ultraviolet radiation sensors onboard a series of polar-orbiting satellites to provide total column and profile ozone products with monthly frequency. Retrievals include those from first- and second-generation SBUV sensors (algorithm version 8.7) and from Nadir Profilers included in the Ozone Mapping and Profiling Suite (OMPS NP, algorithm version 2.8). We use the 2000-2020 portion (21 years) of the MOD, a) to focus on the time period after which the decline of ozone in the upper stratosphere stopped (Newchurch et al., 2003; WMO, 2007), b) for consistency with other studies, and c) to avoid issues with coverage gaps in the earlier parts of the record. Indeed, the MOD provides nearly global spatial coverage but suffers from substantial gaps in its temporal coverage. For instance, no reliable measurements were available from April 1976 through November 1978 and following large volcanic eruptions that injected aerosols in the stratosphere and affected the retrieval (e.g., mid-1991 through 1993, see Bhartia et al. (1993)). Early portions of the dataset are also unsuitable for trend analysis due to partial instrument failure (May 1970 to April 1976). Note that among existing merged datasets, the SBUV record provides the densest and most spatially uniform sampling over the 2000-2020 time period (Tummon et al., 2015). Moreover, the MOD record merges data from similar SBUV instruments, and its sampling and retrieval characteristics are more homogeneous over the time period we use than other merged records that rely on more varied data sources.

Both total column and profile ozone products used in this study have 5-degree horizontal resolution and are zonally averaged to minimize instrumental uncertainty. The ozone profile product provides ozone amounts in 7 layers (DU/layer) spanning the mid-to-upper stratosphere, with layer edges near $25, 16, 10, 6.4, 4, 2.5, 1.6,$ and $1$ hPa, or approximately $23, 27, 31, 35, 40, 46, 50,$ and $54$ km near the equator. Data across these layers have accuracy suitable for long-term trend analysis and compare well with ozone records from other space-borne and ground-based instruments (within $5\ \%$, Bhartia et al., 2013; McPeters et al., 2013). Data outside the $25 - 1$ hPa range tend to be heavily influenced by the a priori ozone climatology used in the retrieval algorithm (Bhartia et al., 1996) and are therefore not relevant to this analysis (see Methods section). Total column ozone data lie within $1\ \%$ of the ground-based instrumental record (McPeters et al., 2013).





## 2.4 Merged satellite trend estimates

We use trend estimates from the SPARC/IO3C/GAW Report on Long-term Ozone Trends and Uncertainties in the Stratosphere (LOTUS, Petropavlovskikh et al., 2019) to provide context for trends derived using the SBUV record alone. The homogenized satellite record used in the LOTUS report includes six merged datasets: the SBUV MOD (v8.6, Frith et al., 2014), the SBUV cohesive dataset from NOAA, the Global OZone Chemistry And Related trace gas Data records for the Stratosphere (GOZ-CARDS v2.20, Froidevaux et al., 2015), the Stratospheric Water and Ozone Satellite Homogenized database (SWOOSH v2.6, Davis et al., 2016), the SAGE-OSIRIS-OMPS dataset corrected for sampling effects (corr-SAGE-OSIRIS-OMPS, Damadeo et al., 2018), and the SAGE-CCI-OMPS dataset from ESA (Sofieva et al., 2017).

The LOTUS trend estimates are based on multi-linear regression and include proxies to correct for the effects of a range of known climate oscillations (El Niño Southern Oscillation, ENSO; the QBO, the 11-year solar cycle, the Arctic and Antarctic Oscillations, the North Atlantic Oscillation), as well as changes in the stratospheric meridional circulation, tropopause height, stratospheric sulfate aerosol abundances, and changing long-term trends in chemically reactive halogens (Steinbrecht et al., 2017). A comprehensive list of references for these proxies is provided in the LOTUS report (Petropavlovskikh et al., 2019). Trends are calculated as the unweighted average of trends from each of the six merged datasets. Trend uncertainties account for the degree of dependency between the datasets and the propagation of errors in the regression model (see section 5.3.2 in Petropavlovskikh et al., 2019). In this study we use trend estimates for the 2000-2020 period, shown in Figures 8 and 9 for three broad latitude bands.

## 3 Methods

### 3.1 Synthetic SBUV observations for the pre-industrial era

In order to quantify the impact of the satellite retrieval process on the observed variability of stratospheric ozone, we create synthetic observations by applying the SBUV kernels to the ESM4.1 pre-industrial control run. As discussed in the previous section, the model run itself is used as reference, and the "kernelized" run is used as equivalent observations of the atmosphere had satellites equipped with SBUV sensors flown during the pre-industrial era.

### 3.1.1 Averaging kernels for synthetic observations

Ozone profiles retrieved by remote sensing are not perfect measurements of the ozone concentrations in each layer of the atmosphere with independent errors, but rather best estimates made given measured radiances and some prior knowledge about the state of the atmosphere. Retrievals are an estimate of some smoothed function of the actual state, with errors that are correlated between different altitudes. As described in Rodgers (1990, 2000), the retrieved ozone profile $\hat{x}$ is related to the true ozone profile $x$ and an a priori $x_a$ used in its retrieval by:

$$\hat{x} - x_a = A(x - x_a) + \epsilon_x$$



where $A$ is the averaging kernel matrix and $\epsilon_x$ represents random and systematic measurement errors. The i-th row of $A$ describes where in the column the information attributed to the i-th vertical level actually comes from. Thus, averaging kernels are peaked functions whose half-widths correspond to the vertical resolution of the retrieval. Two examples for SBUV are shown in Figure 2 (also see Figure 2 in Kramarova et al. (2013a) for more examples). The blue line shows the row of the kernel matrix that corresponds to the $10.1 - 6.4$ hPa layer and the red line shows the same for the $101.3 - 63.9$ hPa layer. The

averaging kernel for the upper level has a peak at the target level, so the concentration attributed to the $10.1 - 6.4$ hPa layer is the weighted average of concentrations between about 2 and 25 hPa, but with the largest weight at the target level. The averaging kernel that is intended to represent the $101.3 - 63.9$ hPa layer has a peak around 50 hPa and exhibits large sensitivity to tropospheric ozone, as discussed in Bhartia et al. (2012). Visualizing the kernels helps to demonstrate the relatively superior quality of SBUV ozone retrievals in the middle to upper stratosphere compared to those in the lower stratosphere; for vertical

levels below 25 hPa, averaging kernels have broad peaks that are significantly shifted away from their respective target levels, indicating that the retrieval for those levels heavily relies on a priori information (Rodgers, 2000). For this reason, as stated in the data section, the analysis with SBUV is limited to the pressure levels $25.45$ hPa through $1.013$ hPa (levels number 9 through 15 in the MOD files) and to total column. Kernel matrices are produced for each retrieval but are only available as monthly averages. We use the monthly averaged kernels for one representative year, chosen to be 2005. For simplicity and to reduce the

computational cost of the study, we only use the SBUV kernels and retrieval algorithm, even though the MOD record includes OMPS retrievals; note that MOD relies at least in part on SBUV until March 2018.

A priori information typically consists of a climatology obtained via independent measurements. For instance, a priori states for the SBUV merged data set come from a combination of independent satellite retrievals and reanalysis data validated by comparison with balloon-borne ozonesondes from the Southern Hemisphere ADditional OZonesondes (SHADOZ) network

(Ziemke et al., 2021). However, no such independent climatology exists when it comes to creating synthetic retrievals using a pre-industrial model run. Synthetic a priori information is therefore created using the model monthly mean state itself, that is to say, the averaging kernels are applied to the departure of the model state from its monthly climatology. The synthetic a priori is therefore unbiased by construction, and the synthetic retrievals represent a best-case scenario. As a consequence, variability in the synthetic retrievals may be underestimated, but this approach has the added benefit of isolating the effects of the SBUV

kernels on the retrieved variability in stratospheric ozone from any effect of the a priori. The relationship between the synthetic retrieval and the simulated ozone (the reference, or 'true' quantity) is written as:

$$\hat{x} = \bar{x} + A(x - \bar{x}) + \epsilon_x$$

where $\bar{x}$ denotes the monthly averaged zonal mean simulated ozone profile. As in Rodgers and Connor (2003), the simulated ozone profile is first interpolated onto the coarser SBUV vertical grid and then convolved with the SBUV averaging kernel

matrix to produce synthetic retrievals. An example of a synthetic retrieval is shown in Figure 2 to demonstrate the effect of the kernels on the vertical ozone profile. The synthetic profile appears smoother than the true profile (hence the name 'smoothing' errors) and is closer to the a priori than the true profile. Below 25 hPa in the lower stratosphere, the averaging kernels become



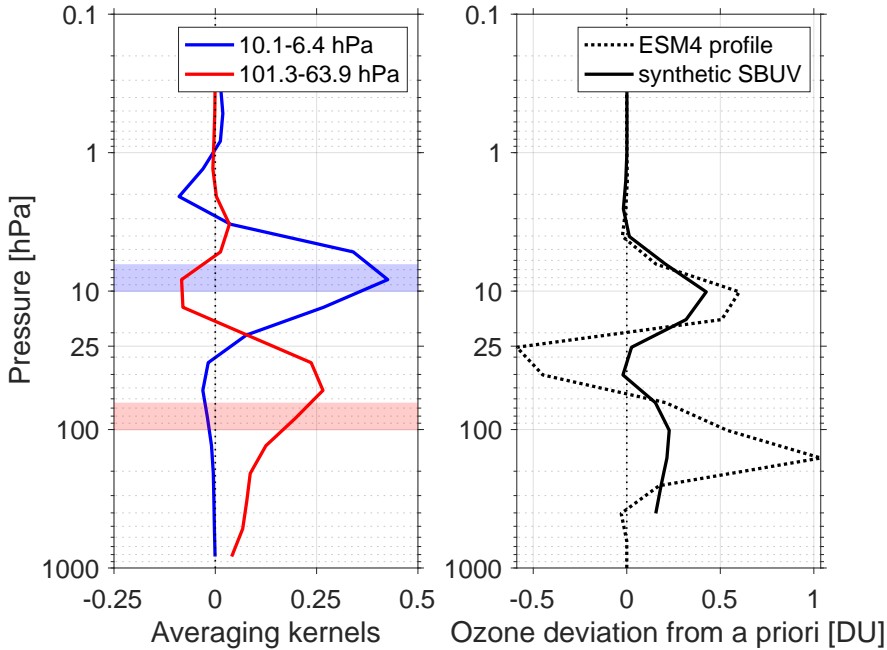

**Figure 2.** (left) Two rows of the averaging kernel matrix and their corresponding SBUV layers (see legend) for $47.5°$N in July 2005, normalized using the SBUV a priori as in Kramarova et al. (2013a) (their equation 3). (right) One example of a model profile and its synthetic SBUV counterpart for the same time and location, shown as deviations from the model a priori to emphasize vertical structures.

broader and the retrieval's quality degrades accordingly, hindering the accurate retrieval of large vertical gradients in ozone concentration.

### 3.1.2 Considerations of known sources of errors

Random errors (e.g., instrumental noise) in retrieved SBUV profiles are of order 1 % (N. Kramarova, personal communication), but since retrievals are averaged zonally and monthly across ∼1000 profiles, random errors in the MOD product are neglected, consistent with Kramarova et al. (2013b). This substantial sample size also allows us to ignore the effects of the spatial sampling of SBUV platforms: monthly zonal means in the pre-industrial model are equated with monthly zonal means in the SBUV dataset.

Several non-random error sources in $\epsilon_x$ produce latitude- and month-dependent biases; these include errors in a priori profiles as well as calibration-related errors in atmospheric attenuation estimates (so-called "N values", see Bhartia et al.,





2013). However, errors that normally affect a priori states are not relevant to this study since the a priori states are known with certainty (see section 3.1.1). In addition, since we analyze decadal trends, data are de-seasonalized and calibration errors that

normally affect the seasonal cycle can thus be ignored. Other, more sporadic sources of errors, including polar mesospheric clouds, volcanic aerosols, ash, and smoke, are also neglected. Accounting for errors resulting from the merging of different SBUV records and biases from non-uniform temporal sampling (orbital drift) is also challenging because of the nature of our analysis. Therefore, such errors are also neglected.

Given these considerations, the error term $\epsilon_x$ is considered negligible. What differs between the synthetic retrievals and the

simulated ozone is their respective evolution over the course of the 500 years of simulation—any differences being attributable to the averaging kernels. The impact of these differences on ozone variability is shown in Figure 3: decadal to multi-decadal variability is misrepresented throughout the upper and middle stratosphere, with errors in integrated power frequently exceeding 25 % and occasionally approaching or exceeding 100 %. Unless these errors are explicitly accounted for, they will affect the apparent variability about SBUV trend estimates, and by extension the degree of statistical significance associated with those

trend estimates, in a general signal-to-noise ratio sense. Regions in blue indicate where the statistical significance of trends will be overestimated, since internal variability is underrepresented there. Regions in red indicate where statistical significance is underestimated; for instance, between $0 - 2.5°N$ in layer $16.4 - 10$ hPa, the variability about trends is inflated because the averaging kernels assign variability from adjacent layers (see Figure 5a) to this layer.

### 3.1.3 Removal of known modes of variability

As in other studies (e.g., Steinbrecht et al., 2017; Petropavlovskikh et al., 2019), an attempt is made to remove the contributions of known modes of variability to the time series of ozone in order to better isolate the internal variability. Several known modes of variability are frequently considered: the seasonal cycle, ENSO and the QBO on interannual time scales, and the solar cycle of irradiance on decadal time scales. Solar irradiance is constant in the pre-industrial run, so no removal of that variability is needed. The seasonal component of variability is modeled as the long-term average ozone abundance for each month of the

year and subsequently removed. Remaining ozone variations are then modeled using the following regression model similar to that used by Stolarski et al. (1991):

$$\Delta O_3(t) = \alpha \text{QBO}_1(t) + \beta \text{QBO}_2(t) + \gamma \text{ENSO}(t) + \epsilon(t)$$

where $\alpha$, $\beta$, and $\gamma$ are parameters determined using regression analysis. Unexplained ozone variability is represented by $\epsilon(t)$. $\text{QBO}_1(t)$ and $\text{QBO}_2(t)$ are the two leading empirical orthogonal functions (EOFs) of the de-seasonalized zonal mean

monthly zonal wind between 10 hPa and 70 hPa (see Figure A1), as in Wallace et al. (1993). Following Oman et al. (2013), $\text{ENSO}(t)$ is based on the NOAA Oceanic Niño Index (ONI) and captures variations of (modeled) sea surface temperature in the Niño 3.4 region (5°S-5°N, 170°W-120°W in the Central Pacific Ocean, see NOAA Physical Science Laboratory: https://psl.noaa.gov/data/timeseries/monthly/NINO34/, last access 6 June 2024) lagged by two months as in Randel et al. (2009). In the pre-industrial model run, $\text{QBO}_1(t)$ and $\text{QBO}_2(t)$ account for 68 % of the normalized variance of the zonal wind

time series. By construction, the correlation between the two EOFs is exactly zero. Thanks in part to the substantial length of



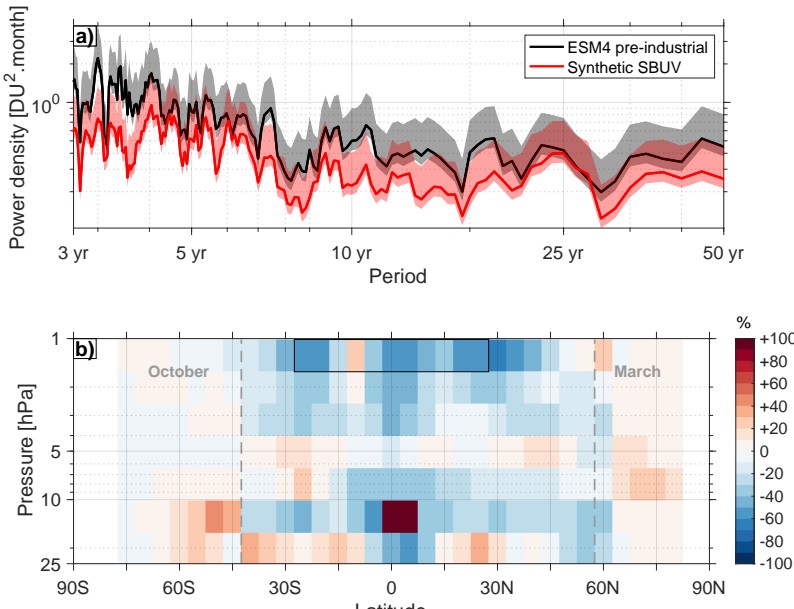

**Figure 3.** Effect of the SBUV retrieval process on decadal to multidecadal ozone variability visualized a) locally, for the topmost SBUV layer, $1.6 - 1$ hPa, between $27.5°$S-$27.5°$N (see the black box in panel b), and b) globally, as the percent relative error in integrated power density between the pre-industrial run and the synthetic SBUV time series for periods 10-60 years. Results for only October or March are shown in the high-latitude regions where SBUV data are not available year-round.

the pre-industrial run, the correlation between the QBO EOFs and ENSO($t$) is very small (less than 0.05), yielding virtually independent regression terms.

The zonal wind data in Figure A1 clearly show that the model's internally generated QBO is far from realistic: the weak QBO-like signal is confined to lower pressures, is much more prone to disruption, propagates vertically at more variable
rates, and exhibits higher frequencies than the observed QBO. Given these differences, the appropriateness of the traditional EOF-based QBO removal could be called into question, but the model does produce a semi-periodic, downward-propagating signal reminiscent of the QBO. Such signals produce large retrieval errors ($1 - 6$ % in observations) when convolved with the averaging kernels (Kramarova et al., 2013a). Since errors of this nature are the focus of this study, we use the traditional EOF-based approach to removing QBO variability. The QBO removal is performed after the simulated ozone is convolved
with the SBUV averaging kernels, to mimic the residual errors produced by the removal of the QBO in SBUV observations.

While the use of EOFs allows us to account for variability due to changes in the amplitude and phase of zonal wind oscillations, it does not account for variability due to changes in the frequency of the oscillations. This is slightly problematic for the removal of the QBO signal in ozone since its frequencies differ between the equator (where the EOFs are extracted




from) and higher latitudes (Tung and Yang, 1994). Nevertheless, the average contribution of the QBO to near-equatorial ozone

variability is estimated to be about 0.5 % in the model (much smaller than the estimated 10-20 % in observations, see Brunner et al. (2006)) and is expected to be even smaller at extratropical latitudes, where the atmospheric jets produce large variability. The EOF-based removal of QBO variability is therefore considered sufficient for the purposes of this study.

The methods discussed in this section provide a set of synthetic SBUV retrievals in the absence of anthropogenic forcings and without known climate oscillations. We use this synthetic record to quantify the internal variability of the climate system

relevant to stratospheric ozone, so as to estimate the lead time required for long-term trends in ozone to emerge in observations against the backdrop of internal variability. For simplicity, we assume that the modern climate system produces the same internal variability as during the pre-industrial era. However, while this assumption has been widely used in the analysis of climate model output (e.g., Houghton et al., 2001; Hegerl et al., 2007; Deser et al., 2012), evidence exists that it may not hold for some variables in climate models (e.g., Schär et al., 2004; Scherrer et al., 2005; Rodgers et al., 2021), though different

models exhibit different responses in their internal variability under forced change (Maher et al., 2021). Therefore, we suggest caution in interpreting statistical confidence.

## 3.2   Time of emergence of linear trends

Several methods have been employed in the past to calculate the time of emergence of linear trends. Emergence has sometimes been said to occur when the ratio of the signal (trend) to the noise (internal variability) is large or exceeds a pre-determined

threshold (Madden and Ramanathan, 1980; Wigley and Jones, 1981; Giorgi and Bi, 2009; Hawkins and Sutton, 2012). Such methods do not provide the formalism needed for a measure of statistical confidence that links the robustness of the results to the choice of threshold to exceed. Another method (Mahlstein et al., 2011) defines the time of emergence as that when a statistical test deems an observation unlikely to belong to a distribution that represents an unperturbed climate state. A problem with this method, however, is that it does not account for autocorrelation in the time series, which means that the time of

emergence of time series that have nonzero autocorrelation is underestimated. Other methods do take autocorrelation into account, for instance that of Li et al. (2017) in which emergence occurs when the confidence interval about a cumulative trend excludes zero. In their framework, the confidence interval is described analytically based on the work of Thompson et al. (2015) and accounts for autocorrelation. However, the method requires the residuals representing internal variability to be normally distributed. This is generally not the case for stratospheric ozone: after removing the seasonal cycle and the contributions from

ENSO and the QBO, the residuals oftentimes exhibit absolute skewness greater than 0.5 and large positive excess kurtosis (see an example in Figure 4 panel b).

### 3.2.1   Definition of the time of emergence

For these reasons, the time of emergence is quantified using the method of Rivoire et al. (2024) (R24 hereafter), which provides nearly identical results to Li et al. (2017) but can handle non-normally distributed residuals. R24 found empirical evidence that

the method of Li et al. (2017) may still be used when residuals are non-normally distributed, but without any formalism to prove this generally, we consider the method of R24 to be a safer choice. The general principle behind the method is to compare an

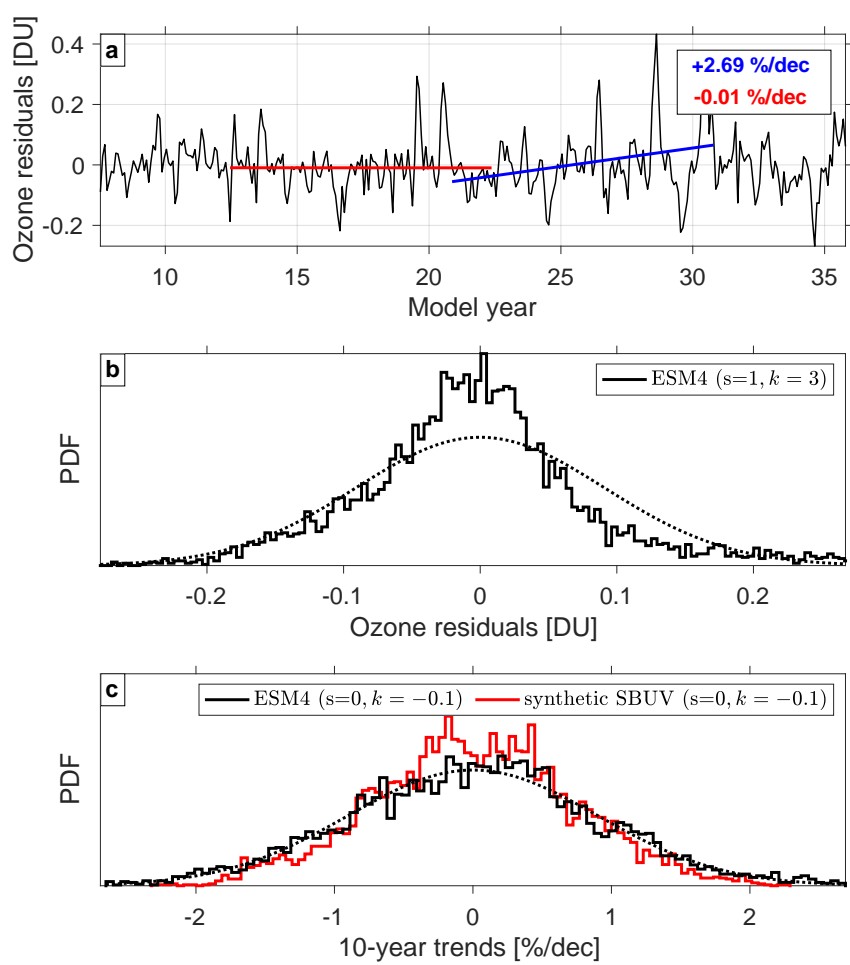

**Figure 4.** a) Sample time series of pre-industrial ozone residuals at $2.5 - 1.6$ hPa, $42.5°$ S, with 10-year trend examples calculated in percent of the mean ozone value at that location, $\sim 4.5$DU. b) Distribution of ozone residuals for the 500 years of simulation, and c) corresponding reference distribution of 10-year unforced trends. Dashed lines show Gaussian distributions with moments identical to those from the model distributions, for reference. The skewness and excess kurtosis of the distributions are provided in the legend.

observed trend of interest, denoted $b$, to a reference distribution of unforced trends established by sampling naturally occurring trends in the control run (see Figure 4 panel a and c for an example). The time of emergence, denoted $y$, is the length of the sampling window such that the magnitude of the observed trend $b$ is greater than trends in the reference distribution at





least $c = \frac{c_d + 100}{2}$ % of the time, with $c_d$ the chosen statistical confidence threshold, in percent. In the rest of the paper we use $c_d$=95 % as the threshold for statistical significance and use a two-sided test. Though the use of this particular threshold is conventional, we emphasize its arbitrary nature and caution against over-interpreting its significance.

Since the time of emergence $y$ is calculated using a trend distribution sampled directly from a model run, it neglects the effects of observational uncertainties. In other words, $y$ is only valid if one possesses omniscient knowledge of the atmosphere– we refer to it as the "ideal" time of emergence, see Figure 5. R24 show that observational uncertainties can introduce large errors in $y$, and they provide the formalism to correct $y$ accordingly. In our case, the correction for $y$ accounts for the effects of the vertical redistribution of ozone variability by the averaging kernels (see Figures 2 and 3). The corrected time of emergence is denoted $y^*$ and is calculated using a reference trend distribution sampled from the synthetic SBUV retrievals introduced earlier, rather than the model run itself. As a result, $y^*$ quantifies the time of emergence of ozone trends *as seen by SBUV platforms*.

### 3.2.2 Notes on the realism of the pre-industrial control simulation

Since emergence is defined based on the representation of internal variability from a model, the realism of the model comes into question. On the one hand, the time of emergence $y$ is subject to the realism–or lack thereof–of the model: misrepresentations in the magnitude or frequency spectrum of internal variability yield errors in the time of emergence. Though we provide an analysis of said realism to the extent possible (section 2.1), quantifying these errors is inherently difficult given the absence of pre-industrial reference for observed variability. On the other hand, R24 showed that the adjustment for the effect of observational uncertainties on the time of emergence, when expressed as a relative error ($\frac{y^* - y}{y}$), is accurate even if the model dramatically misrepresents internal variability. For this reason, we quantify the effect of the SBUV kernels on the time of emergence as a relative error (see section 4.2).

## 4 Results

### 4.1 Time of emergence

Figure 5 provides estimates for the time of emergence ($y$, in years since 2000) with 95 % confidence for both a spatially homogeneous 1 %/decade trend (panel b) and for the 2000-2020 trends from CCMI1 (panel c). To put this into context, panel a) shows the average ozone abundances and their standard deviation in the pre-industrial run. The spatially homogeneous trend serves to illustrate the effects of internal ozone variability on the time of emergence. Except for the lowermost stratosphere, a general correlation is seen between the time of emergence and ozone variability as quantified by its standard deviation: the largest values are found in the tropical middle stratosphere and polar regions. Departures from this correlation arise where the spectrum of variability includes non-Gaussian behavior and is not well described by the standard deviation alone.

According to our method, and based on ESM4.1 as a reference for internal variability, trends associated with the modeled evolution of ozone under the Montreal Protocol may not emerge until the second half of the century in the heart of the midlati-





tude ozone layer (panel c). In the upper stratosphere (above 10 hPa), chemistry-climate model trends for 2000-2020 are largely distinguishable from ESM4.1's natural variability with ∼20 years of model record, in agreement with previous literature showing significantly increased ozone abundances (Gillett et al., 2011; Arblaster et al., 2014). Lower down, especially in the middle stratosphere (10 − 30 hPa), our method finds that trends have generally not emerged yet. In the midlatitudes (35 − 60°N/S),

trends may require decades of additional record to emerge, owing in part to large variability and in part to small trends. This finding is generally consistent with existing literature, which finds little to no significance to trends in the region (WMO, 2022, and references therein). Over the Arctic, we find that the emergence of positive trends could occur by 2030, compared with literature expecting visible signs of recovery there in the mid-2040s. However, since our results are based on the analysis of a single model, we urge caution: previous analyses have concluded that large dynamical variability in this region precludes the

detection of recovery until the late 21st century (WMO, 2022). It is possible that the ozone interannual variability in ESM4.1 still lacks realism despite recent improvements (Horowitz et al., 2020, their Figure 11).

For vertically integrated ozone (panel d), despite the relatively small variability in the tropical region, emergence occurs later there because trends are small. Trends associated with chemical loss over the polar caps in spring are isolated using October- and March-only time series in the Southern and Northern Hemispheres, respectively. Though ozone loss is largest

over Antarctica and is therefore expected to exhibit the clearest signs of recovery (Newchurch et al., 2003; Yang et al., 2008; Charlton-Perez et al., 2010), the analysis predicts that trends for October and March only emerge after ∼40 years near the strongest trends (65°N/S, not shown), owing to large variability. Closer to the poles, October and March trends become very small and therefore take much longer to emerge.

Some caveats should be kept in mind: the time of emergence is based on a representation of unforced ozone variability that

excludes the effects of ozone-depleting substances or volcanic aerosols (see Figure 1). As a result, time of emergence estimates provide lower bounds for the timing of the detection of recovery (as long as the linear trend approximation holds). Further, in previous literature, statistically significant trends associated with the recovery of the ozone layer are based on more than just ozone abundances; they include trends in other metrics, such as the minimum in 15-day average total ozone, maximum in 15-day average 220 DU ozone hole area, minimum in 15-day average ozone mass deficit, and the duration of the ozone hole

(e.g., Tully et al., 2020). It is possible that these metrics provide more detectable trends than ozone abundances alone. Lastly, we highlight that quantifying the emergence of a trend *vs.* that of an epoch difference may yield different results.

### 4.2 Effect of the SBUV retrieval on the time of emergence

Results established so far apply to the "model world" only and do not factor in the limitations of observing platforms that make up the historical record. The synthetic SBUV record from section 3 provides an example of the effect of observational

limitations on the time of emergence. Figure 6 shows those limitations as the relative difference between the retrieval-adjusted time of emergence $y^*$ and the ideal time of emergence $y$. Large errors throughout the middle and upper tropical stratosphere are reminiscent of the patterns of variability associated with the QBO, despite the prior removal of QBO variability (Section 3.1.3). This result arises because the convolution of the SBUV kernels with the ozone data affects the representation of the QBO (Kramarova et al., 2013a), but does not affect the wind field that is used to perform the QBO removal. The $16 − 10.1$





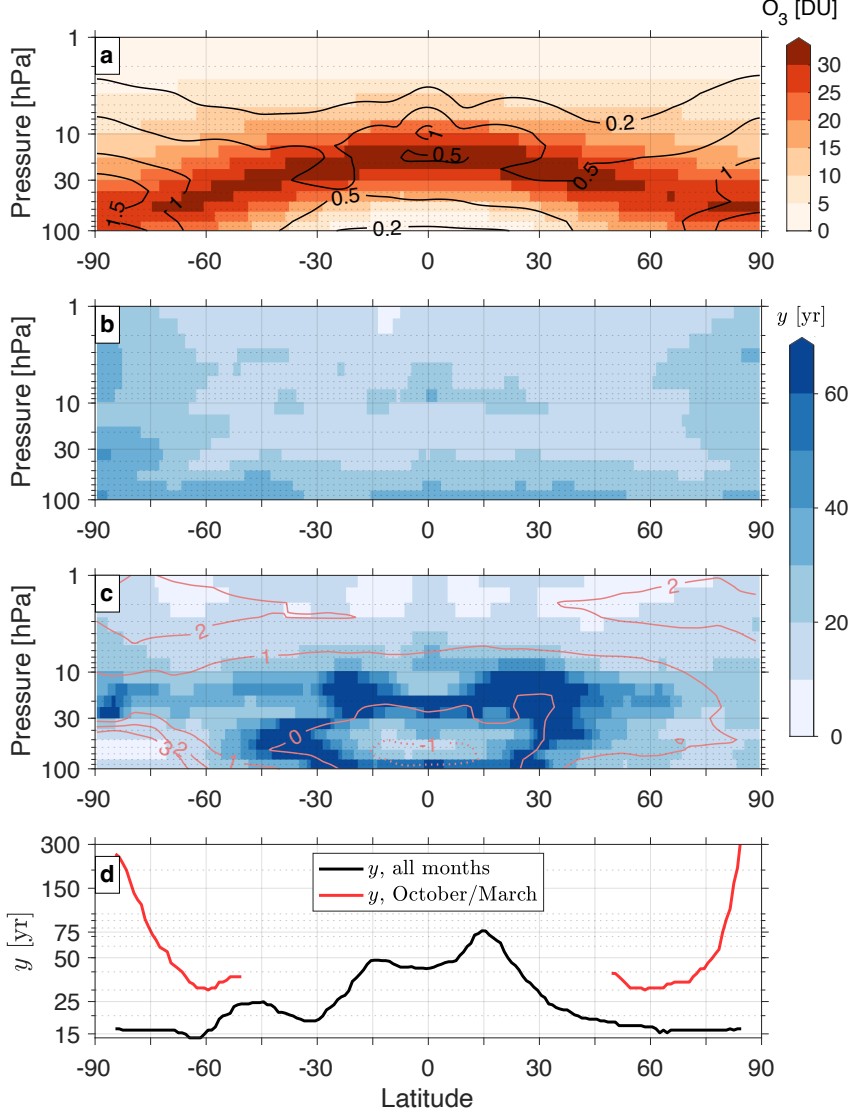

**Figure 5.** a) Average ozone abundance in the ESM4.1 pre-industrial run (in DU, colors) and standard deviation of the de-seasonalized time series (black contours, same units). b) Ideal time of emergence for a spatially homogeneous 1 %/decade trend, in years with contours every 10 years. c) Ideal time of emergence for 2000-2020 trends in CCMI1 ref-C2 (shown by overlaid red contours, %/decade). d) Ideal time of emergence for 2000-2020 trends in CCMI1 ref-C2 total column ozone. Results for October-only (March-only) are shown in the Southern (Northern) Hemisphere.

hPa layer exhibits a large bias at all latitudes, arising in part from the relatively large ozone variability in that layer (refer to Figure 5a). Errors are also large over the polar caps, in connection with the unavailability of SBUV retrievals during polar




night. Interestingly, errors are opposite in sign between the two polar caps at some levels, indicating opposite effects due to the SBUV sampling and retrieval process there. Errors are slightly larger over the Southern polar cap than the Northern, which is attributable to slightly larger internal variability in ESM4.1 in the Antarctic (as shown in Figure 5a). This result is more

specifically attributable to the shape of the power spectrum of internal variability in ESM4.1 around periods corresponding to the time of emergence. The largest negative errors found near the equator just below 10 hPa are explained by the unusual overestimation of internal variability attributable to the SBUV retrieval process shown in Figure 3b. At all latitudes, the error pattern is qualitatively unchanged whether trend magnitudes are homogeneous (panel a) or not (panel b), indicating that our method provides generally applicable understanding of the deficiencies of the SBUV ozone retrieval for detecting trends.

For total column ozone, the retrieval-adjusted time of emergence estimates are very close to the ideal estimates (Figure 6c). This is consistent with the result in Kramarova et al. (2013a) (their Figure 6) that combining SBUV layers improves the accuracy of ozone retrievals.

### 4.3 Effect of the SBUV retrieval on vertically resolved trend estimates

Retrieval methods based on averaging kernels have previously been said not to affect trend estimates (Petropavlovskikh et al.,

2019). The rationale behind this statement is straightforward: since the retrieval process remains unchanged over time, errors attributable to it are also constant over time (at least in a long-term sense), and trend estimates are therefore unbiased. However, this rationale fails to account for the vertical redistribution of information by the kernels. As trends affect the true profile ($x$ in section 3.1.1), the difference between the true profile and the a priori ($x - x_a$) also contains a trend signal. Once the kernel matrix is convolved with ($x - x_a$), the trend signal is redistributed vertically. As a result, trend estimates at each vertical level

are affected by the trend signal from adjacent levels in a manner defined by the shape of the averaging kernels. This is true of any profile, including the hypothetical case of a vertically uniform profile in the context of trends expressed in percent per unit time, but errors can be especially large when trends are vertically heterogeneous. Figure 7 shows such errors for a few idealized trend profiles representing likely ozone increases in the mid-to-upper stratosphere (Godin-Beekmann et al., 2022). The SBUV retrieval process produces a noticeable smoothing of the profile and misrepresents the vertical position and magnitude of local

maxima. Errors as large as 100 % and displacements as large as 6 km are found, highlighting the importance of accounting for the retrieval process in estimating trend uncertainties. Negative correlation coefficients between layers in the averaging kernel matrix can yield negative trends even when the true trends are only positive.

The results suggest that trends in the middle and lower stratosphere may be affected by the presence of trends in the upper stratosphere. This is particularly relevant in the context of analyses that show negative trends in the lower stratosphere (e.g.,

Ball et al., 2018). While the errors near 25 hPa may seem small, they may induce large errors in the time of emergence. Altogether, this analysis shows why trends should be analyzed as vertical profiles rather than at individual vertical levels.

### 4.4 Smallest detectable trends

When trend estimates are in disagreement across model simulations or observational records, the method from R24 can be used to determine the range of trend magnitudes that should be distinguishable from internal variability in the first place.



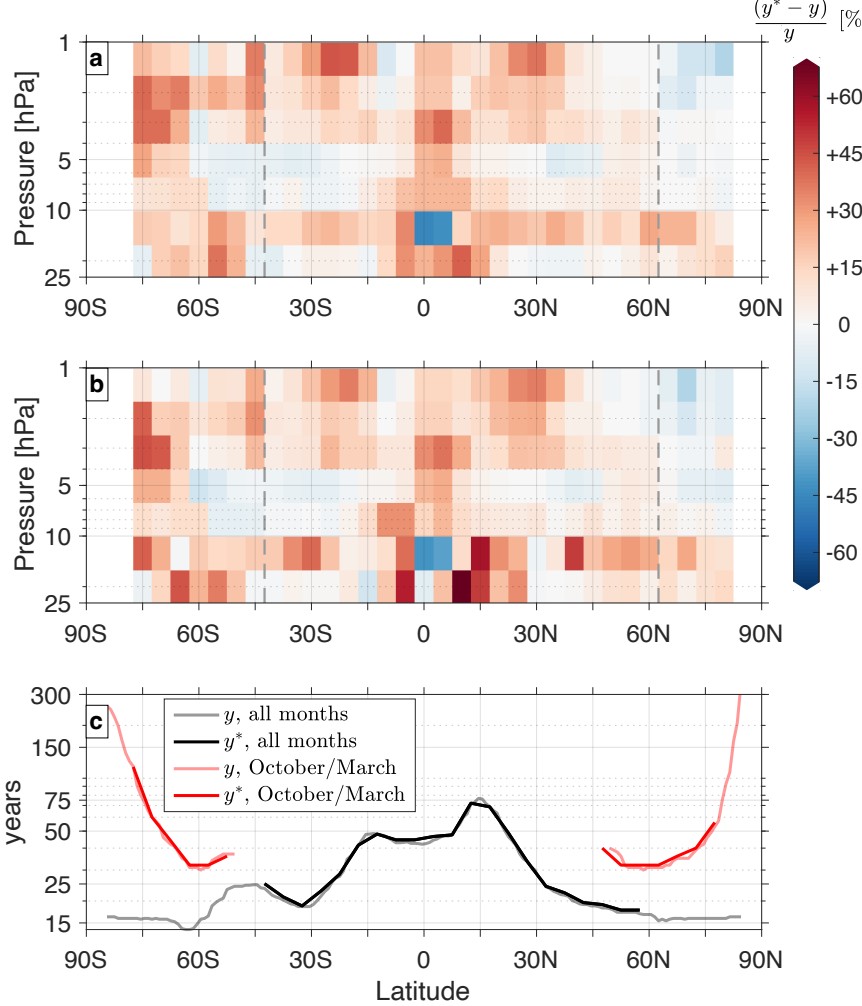

**Figure 6.** Relative error on the time of emergence due to the SBUV kernels for a) a spatially homogeneous 1 %/decade trend and b) the CCMI1 trends from panel c) in Figure 5. Dashed gray lines denote the latitudes poleward of which the SBUV retrieval is not available year-round. c) Ideal time of emergence ($y$) and kernel-adjusted time of emergence ($y^*$) for 2000-2020 CCMI1 trends in total column ozone.

Using the time of emergence $y$ as the length of the sampling window used to calculate trends from the control run (or its synthetic counterpart), $b$ is the smallest detectable trend at confidence level $c_d$. Trends with magnitude smaller than $|b|$ are not distinguishable from internal variability at the chosen degree of confidence. The magnitude of $b$ provides an "envelope of undetectability", shown in Figure 8 for a few key latitude bands, assuming omniscient/model knowledge. Simulated and observed 2000-2020 trends in the upper stratosphere (above 10 hPa) largely lie outside the range of undetectable trends, 390   indicating that the trend estimates have emerged from internal variability there. Interestingly, the inter-model spread lies outside





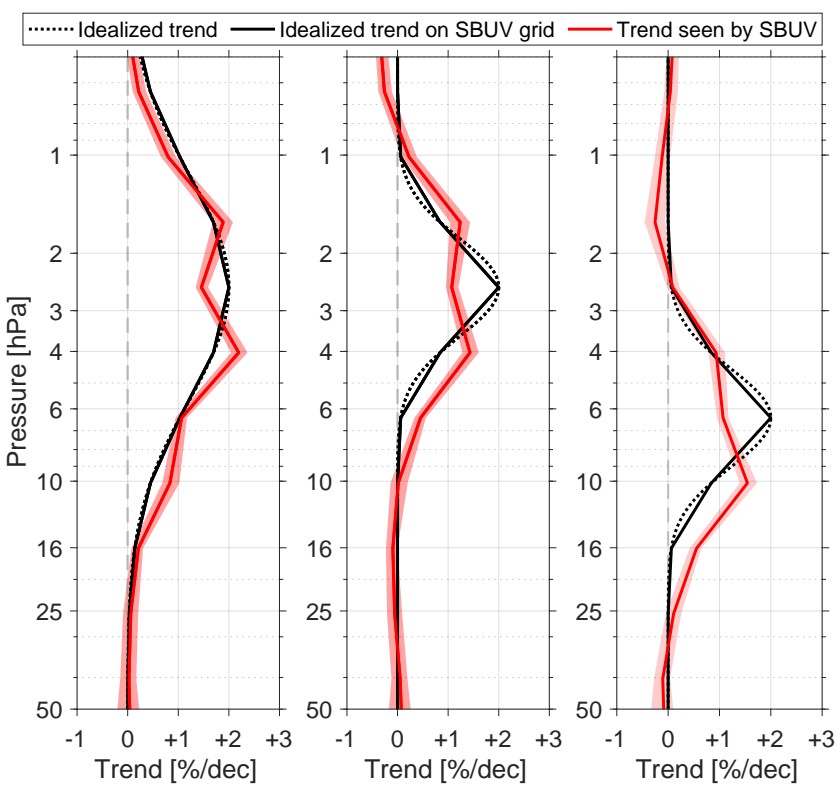

**Figure 7.** Idealized trend profiles (black) and corresponding profiles as would be observed by the SBUV retrieval algorithm at $42.5°$N (red). The red shading shows the interquartile range for trends estimated using 24 years of data (roughly the length of the record since 2000).

the envelope only above $3 - 5$ hPa. Below 10 hPa, mean trends are located within the range of undetectability, which means that internal variability is so large that 21 years of data are not sufficient to ascertain whether ozone is increasing or decreasing with 95 % certainty there. Further, simulated and observed trends within the heart of the ozone layer (see Figure 5a) are currently not detectable with 95 % statistical confidence, according to the data sets used, and again, assuming omniscient knowledge.

The same analysis is performed on the synthetic SBUV control simulation to obtain the envelope of undetectability accounting for observational uncertainties. Figure 9 overlays this SBUV envelope and that given by the unaltered model run, that is, the ideal envelope of undetectability (from Figure 8). Overall, SBUV envelopes are optimistic (narrow), because the SBUV retrieval process tends to reduce the true internal variability (in this latitude-band-average sense; compare to Figure 3) or affect its spectrum in a way that artificially decreases the time of emergence. The differences between the true and SBUV envelopes (Figure 9) are most noticeable in the tropical region, where the SBUV retrieval yields errors up to 25 %. Note that averaging





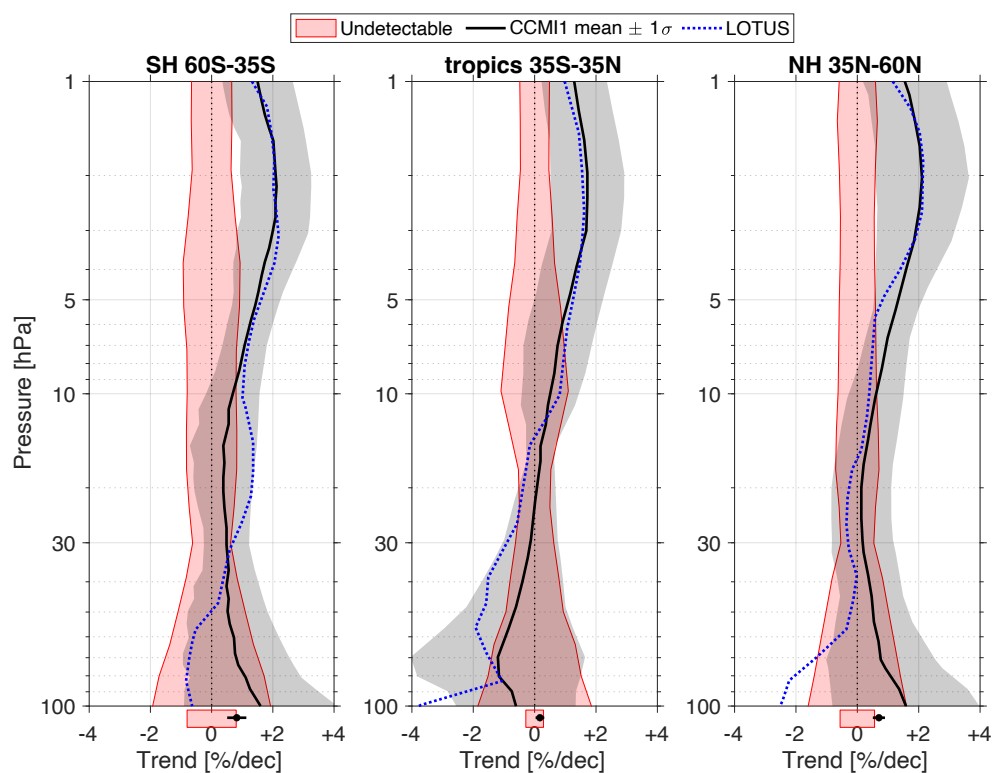

**Figure 8.** Simulated 2000-2020 ozone trends from CCMI1 and the corresponding range of undetectable trends at the 95 % confidence level. Results for total column ozone are shown at the bottom outside each panel. Observed 2000-2020 trends from LOTUS are shown in blue for reference.

the smallest trends over latitude bands reduces the SBUV kernel errors–those can be as large as 50 % locally (not shown). As was concluded from Figure 6, the detectability of total column ozone trends is virtually unaffected by the SBUV retrieval.

The SBUV envelope is most relevant to trends estimated using the SBUV record and it provides useful context for comparative studies that include SBUV trends (e.g., Frith et al., 2014; Harris et al., 2015; Tummon et al., 2015; Solomon et al.,
2016; Chipperfield et al., 2017; Steinbrecht et al., 2017; Weber et al., 2018; Ball et al., 2018; Brönnimann, 2022). We draw a comparison with the LOTUS trends, since LOTUS is subject to the limitations of SBUV, among other datasets. The results are qualitatively similar to those in Figure 8: trend estimates above 10 hPa should generally be distinguishable from internal variability, with the caveat that errors from the SBUV retrieval can yield moderate overconfidence where trend estimates are close to the smallest detectable trends. Note that the large differences between SBUV-only and LOTUS trends are likely due to the
heavy reliance of the LOTUS product on Microwave Limb Sounder (MLS) retrievals, which have narrower averaging kernel




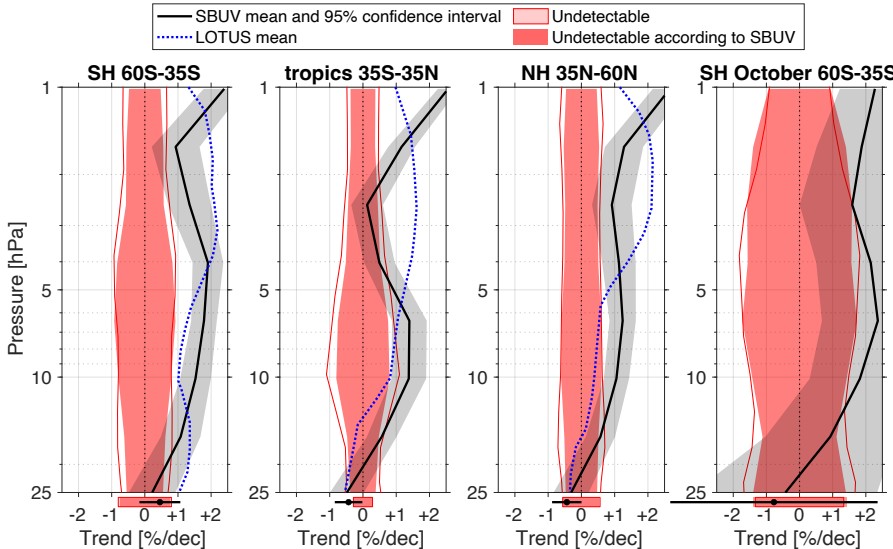

**Figure 9.** SBUV 2000-2020 ozone trends and the corresponding envelope of undetectability at the 95 % confidence level including the adjustment for SBUV kernels. Results for total column ozone are shown at the bottom outside each panel. Observed 2000-2020 trends from LOTUS are shown in blue for reference. The confidence interval about SBUV trends comes from linear regression coefficient estimates.

peaks than those shown in Figure 2, resulting in ∼2.5 km intrinsic vertical resolution in the lower stratosphere (compared to 6 to 10 km for SBUV). Below 10 hPa, most trends are predicted to be indistinguishable from internal variability.

Similar detectability envelopes could be quantified for other nadir sounders, for instance the Global Ozone Monitoring Experiment (GOME, Burrows et al., 1999), Ozone Monitoring Instrument (OMI, Levelt et al., 2006), or Tropospheric Emission
Spectrometer (TES, Worden et al., 2004). Limb sounders are subject to similar errors to the extent that they also rely on averaging kernels, for instance OMPS-LP (Arosio et al., 2018), the Optical Spectrograph and Infrared Imager (OSIRIS, von Savigny et al., 2003), Atmospheric Chemistry Experiment Fourier Transform Spectrometer (ACE-FTS, Walker et al., 2005), Michelson Interferometer for Passive Atmospheric Sounding (MIPAS, Ridolfi et al., 2000), Microwave Limb Sounder (MLS, Livesey et al., 2006), and Stratospheric Aerosol and Gas Experiment III (SAGE III, Cisewski et al., 2014). However, since limb
sounders typically achieve much better vertical resolution than nadir sounders, kernel errors are likely to affect the detectability of trends to a lesser extent.

The results discussed here raise the question: if not now, when will trend estimates eventually become distinguishable from internal variability? To answer this question, Figure 10 shows the magnitude of the smallest detectable trends as a function of the length of the record. In the upper stratosphere (∼1 hPa, Figure 10a), where CCMI1 trend estimates are 1 − 2 %/decade,



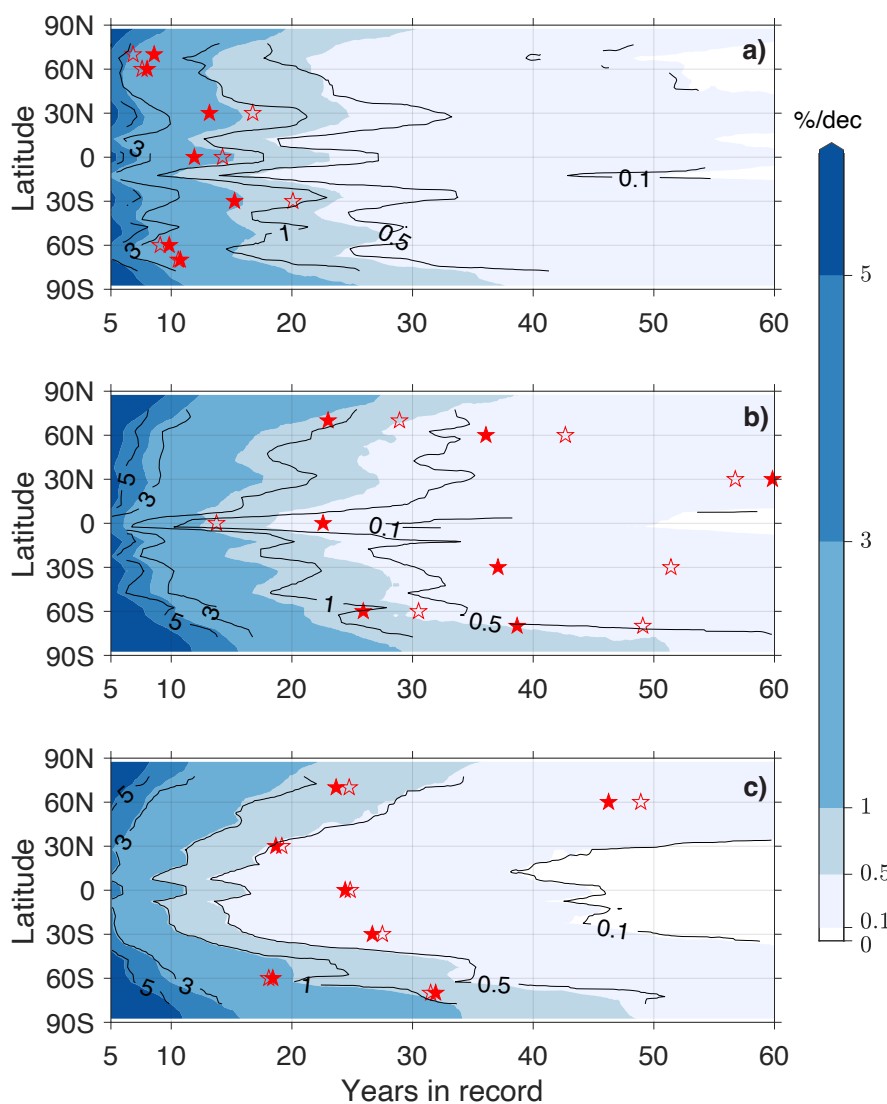

**Figure 10.** Smallest detectable ESM4.1 pre-industrial trends at the 95 % confidence level (shaded) and their SBUV retrieval-adjusted coun-terparts (black contours) given as a function of the length of the available record, for a) $1.6-1$ hPa, b) $25-16$ hPa, and c) the total column. Filled and unfilled red stars indicate the emergence of CCMI1 2000-2020 trends at a few latitudes based on the ESM4.1 smallest detectable trends and their SBUV retrieval-adjusted counterparts, respectively.

the current length of record since 2000 is sufficient to rule out the possibility that the observed increase in ozone is an artifact of internal variability, even when accounting for SBUV kernel errors. This result is in agreement with Figure 5c and despite





the relatively large underestimation of internal variability shown between $60°$S-$60°$N in Figure 3b. Lower down ($25 - 16$ hPa, Figure 10b), CCMI1 trends are about a factor of two smaller and ozone variability is larger; therefore, much longer records are needed. In the Northern midlatitudes, 35 to 60 years of record are needed. Over the Southern polar cap, almost 40 years

of data are needed, increasing to 50-60 years when accounting for SBUV kernel errors. In contrast, the SBUV kernels yield overestimated internal variability near the equator (Figure 3b), and accounting for such errors therefore decreases the length of record needed to detect the CCMI1 trend estimate (the unfilled red star is shifted leftward to earlier years than the filled one). For total column ozone (Figure 10c), the results suggest that CCMI1 trends have emerged, or will soon, except in the Northern midlatitudes and over Antarctica. SBUV kernel errors have little effect on these estimates, consistent with previous

findings (Kramarova et al., 2013a). We emphasize that these results are based on the analysis of a single model (ESM4.1), and that the definition of emergence used here differs from typical measures of statistical significance based on the standard error of regression coefficients. These caveats are important to keep in mind given that observational studies have found signs of recovery in the total column above Antarctica.

## 5 Conclusions

We examined the statistical significance of long-term trends in stratospheric ozone. Statistical significance was tested against the null hypothesis of no *forced change*, rather than the null hypothesis of no change, in order to include information about the magnitude of trends that arise purely as a result of internal variability in the climate system. Using the concept of time of emergence, we showed the potential for large errors in the statistical significance of trends estimated from satellite records. Two factors explain this result: 1) the averaging kernels inherent to the optimal estimation retrieval technique consistently

misrepresent decadal to multidecadal variability in satellite observations, and 2) known modes of variability (ENSO, QBO) interact with averaging kernels, and their removal from the observed record is therefore prone to residual errors.

Further, our analysis showed that by vertically redistributing information, averaging kernels can alter trend magnitudes, in addition to their uncertainties. This result contrasts with the assumption of Petropavlovskikh et al. (2019), who stated that kernels do not affect trends since kernel errors are constant in time. We find instead that the shape of the vertical trend profile is

subject to large errors, with possible vertical displacements in the location of local maxima by several kilometers. Our analysis uses SBUV as an example, but other nadir sounders (e.g., GOME, OMI, TES) are in principle subject to similar effects, with magnitude dependent on their vertical resolution and location of their vertical levels relative to vertical gradients in ozone concentrations and variability. Limb sounders (e.g., MLS, OMPS-LP, OSIRIS, ACE-FTS, MIPAS, SAGE III) may also be affected, albeit to a lesser extent.

We note a few caveats to our study. Time of emergence estimates are subject to the realism of the model we used (ESM4.1), though the relative adjustment to the time of emergence remains accurate. The reference simulation we used excludes the influence of volcanoes, which has been large in recent years. Additionally, the effects of non-uniform sampling and record-merging procedures are also excluded. The time of emergence estimates presented here should therefore be construed as lower bounds (as far as linear trends are concerned). Our time of emergence values may be further underestimated if the internal

variability of the climate system is increasing under forced change (as discussed by Rodgers et al., 2021). In addition, we used an arbitrary 95 % threshold for statistical significance, and quantitative results are of course sensitive to this choice. Lastly, in this study we did not explore derived metrics related to ozone recovery (e.g., size and timing of the ozone hole), which may confer different detection power with respect to long-term trends.

Our results should be interpreted with these caveats in mind, but nonetheless they provide useful context in light of the
varying degrees of confidence placed in trends in recent literature; notably, adding just a few years to the historical record can change the magnitude and even sign of trends in some locations. Based on our results, we recommend systematically accounting for the effects of retrieval algorithms when calculating long-term trends using vertically resolved ozone records, particularly those from nadir-viewing instruments. We further recommend testing the statistical significance of trends against the null hypothesis of no *forced* change, rather than no change at all. Future work in this direction would benefit from long
reference simulations of chemistry-climate models with perpetual year 2000 conditions, to better characterize the variability relevant to the recovery of the ozone layer since then.

*Data availability.* The SBUV merged ozone record is available at https://acd-ext.gsfc.nasa.gov/Data_services/merged/

The climate simulations from GFDL's ESM4.1 model are available from the CMIP6 archive at https://esgf-node.llnl.gov/search/cmip6/

## Appendix A: Representation of the QBO in the pre-industrial simulation

Figure A1 highlights differences between the QBO generated internally by the ESM4.1 pre-industrial model run and the observed QBO (see discussion in Section 3.1.3).

*Author contributions.* LR and ML designed the methods. LR performed the analysis and produced the figures. The text was primarily written by LR with feedback from all co-authors. This project was initiated by ML, JN, and PL (poster titled 'Uncertainty in Ozone Trend Detection' presented at the American Meteorological Society's 2020 Annual Meeting).

*Competing interests.* The authors declare no competing interests

*Disclaimer.*

*Acknowledgements.* We thank Natalya Kramarova (Goddard Space Flight Center) for help with applying the SBUV retrieval. Ozone trend estimates from CCMI1 were kindly prepared by Kleareti Tourpali (Aristotle University of Thessaloniki). We acknowledge discussions





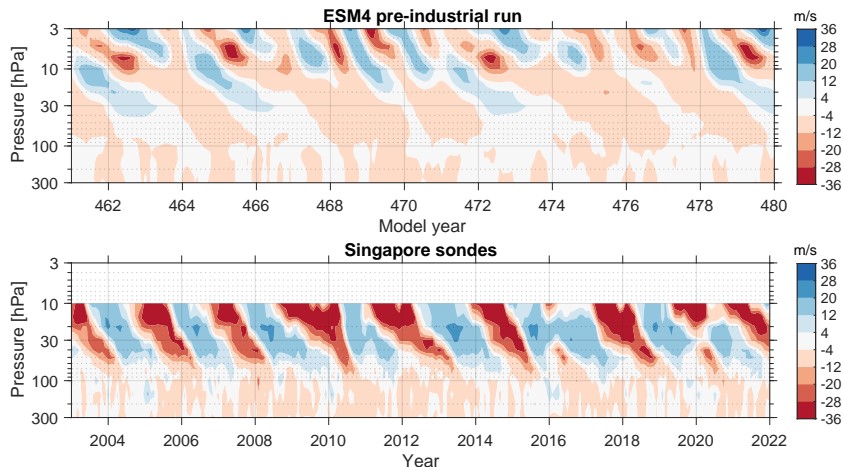

**Figure A1.** Sample time series of the simulated (top) and observed (bottom) de-seasonalized monthly zonal mean zonal wind. Positive values denote westerly wind anomalies.

that shaped the analysis with Irina Petropavloskikh, Peter Zoogman, Aaron Match, Susann Tegtmeier, and Kane Stone, as well as early
contributions to this project by Mary Frances Connors. LR was funded by the William F. Milton Fund and National Aeronautics and Space Administration (NASA) award 80NSSC21K0943. ML was funded by NASA awards 80NSSC21K0943 and 80NSSC23K1005. PL was funded by award NA18OAR4320123 from the National Oceanic and Atmospheric Administration (NOAA), U.S. Department of Commerce. Work at the Jet Propulsion Laboratory, California Institute of Technology, was done under contract with NASA (80NM0018D0004).



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
