# Peer review of "Satellite nadir-viewing geometry affects the magnitude and detectability of long-term trends in stratospheric ozone"

_EGUsphere, 2024_

## Author Comment (AC1)

Response to RC2

Thank you for your careful review—please see our response in blue in the text below.

Summary:

This paper reports on the effect of averaging kernels from nadir-type merged ozone timeseries, here SBUV Mod, on the detectability of long-term ozone trends. Using CCM model data, the uncertainty in trend detection due to internal variability as a function of the record length was investigated. As stated in this study, the uncertainties in the representativeness of modeled internal variability for observations is rather large so that the results from this study should be considered with some caution.

The main issue with the averaging kernels from nadir observations, is that they are quite broad (low vertical resolution) and in the lower stratosphere asymmetric, so that trends at a given altitude have, in some cases major, contributions, from other altitudes.

Overall the paper is well written. My major concern is that neither the abstract nor the introduction clearly state what the scope of this study is and that the results are limited only to a particular subset of satellite ozone profile measurements. One should not use the term retrieval as a synonym for averaging kernels (AK). AKs are used in the retrievals but they are inherent to nadir observation geometry in the UV. So regardless of the algorithm used, all AKs are similar for the same observation geometry. Also a clear distinction should be made early on that this study focuses on nadir-derived ozone timeseries and that other datasets based upon limb observations have narrower AKs and are not investigated here. We've added that distinction in the data Section (2.4), and we've replaced references to the 'SBUV retrieval process' with 'SBUV kernels'.

In many figures panels are missing titles. This would make the figures more readable even without checking the figure caption. This applies to Figs. 1. 3, 4, 5, 6, and 10. We've added panels titles in Figs. 1, 3, 4, 5, 6, and 10, as suggested. Where panels were labeled (a, b, c, etc) we've moved the labels to the panel titles to reduce visual clutter.

Specifics:

Paper title: The paper title is quite unspecific. I also think that it is not clear what is meant with "forced" here. I strongly suggest to add "nadir-derived", like: "Detectability of trends in nadir-derived stratospheric ozone". We appreciate the criticism, and understand that the term 'forced' could cause some confusion, especially because the forcing in this case corresponds to the *decrease* in ODS abundances. We could clarify that the analysis is done "in the context of climate variability," but for the sake of keeping the title short, we propose: "Satellite nadir-viewing geometry affects the magnitude and detectability of long-term trends in stratospheric ozone."

l.2: In this paper the focus is on the impact of averaging kernels, which are inherent to the observation type rather than the retrieval algorithm (see comments above). Retrieval algorithms can have different settings that may also impact trends (e.g. uncertainties due to temperature

dependence of ozone cross-sections). Do not use retrieval algorithm as a synonym for averaging kernels (check also thee entire paper) Thank you for pointing this out; as noted above, we have replaced "retrievals" with "kernels" throughout the paper.

l.4: mention here the use of SBUV MOD and that the data is based upon satellite nadir measurements. We prefer to introduce the specific data sets later on to avoid defining acronyms in the abstract. We instead added mention of the use of broad averaging kernels.

l.22: "to decrease lower stratospheric ozone abundances". This is limited to the tropics, a corresponding increase in extratropical ozone is expected. Done.

l.42: "disagreement". Probably better to say that "models, observations, and trends are highly uncertain" Done.

l.42: see also WMO (2022, p. 165) which recommends to be very cautious with trends derived from reanalyses. Done.

l.45: A bit more explanation on the role of internal variability is needed here (see for instance, doi:10.1038/s41612-023-00389-0). Does the determination of internal variability not require an ensemble of model runs rather than using single (or two) model runs as done in this paper. Please discuss. Quantifying internal variability does require a large sample size, as discussed in the reference you provide. In the case of model simulations with time-dependent external forcings (e.g., for historical time periods), exploring the diversity of initial conditions is crucial as external forcings interact with the internal variability, and small differences in initial conditions can yield different future evolutions. The same is not true of the pre-industrial simulation we are using as control (i.e., without external forcings): that simulation captures a stationary system in which the memory of initial conditions decays due to chaotic processes, and the timescale of the simulation is much longer than the dominant variability modes. In other words, the use of multiple model realizations is important when quantifying internal variability during a particular time period, but that is not our case: we focus on the role of internal variability over *any* time period of chosen length. For this reason, we follow the standard practice of using a single, long control simulation (500 years in our case) as reference for internal variability.
We added a short discussion in the first and second paragraphs of section 2.1 to clarify this.

l.46: the first sentence in this paragraph is misleading as it suggests that all these aspects are considered in this study (see earlier comments). We have modified this sentence in accordance with your comment and that of RC1.

l. 108: add some references to the SBUV MOD data, e.g doi:10.5194/acp-17-14695-2017. We added both this reference as well as the earlier Frith et al. 2014 reference (https://doi.org/10.1002/2014JD021889)

l. 133: one should mention here that all LOTUS datasets except for SBUV are derived from limb observations with narrower and less asymmetric averaging kernels (see earlier comment) We have modified this sentence in accordance with your comment and that of RC1.

l.150: What is the vertical sampling of ESM4.1. State it here or earlier when describing model data. Also it may be helpful to provide some typical numbers for the vertical resolution of nadir-type ozone profiles (see, for instance, 10.5194/amt-14-6057-2021), here or before when describing SBUV MOD. The model features 2-3 km resolution in the lower stratosphere; we've added that to the model section. The range of vertical resolution for nadir sounders is relatively wide and spatially dependent; to avoid lengthening this section, we simply state that the SBUV vertical resolution is comparable to that of other nadir records.

l. 169: this has not to do with retrieval quality as this is an inherent physical property from the retrieval using nadir observations (see comments above). Maybe "limits" is better. We have rephrased the sentence to eliminate the term 'quality,' as suggested.

l. 227 (Eq.) earlier you mentioned that the LOTUS regression on the six merged datasets includes AO/AAO, and/or NAO. Not important here? The LOTUS report states that the AO, AAO, and NAO proxies have a negligible impact on trend estimates. We now mention this in Section 3.1.3 where the removal of known modes of variability is discussed.

l. 253: replace "retrieval" by "data". Done

l. 263. Regarding ozone the following paper paper is also relevant here doi:10.1029/98JD00995. We've added both this reference and that to:

Tiao, G.C., Reinsel, G.C., Xu, D., Pedrick, J.H., Zhu, X., Miller, A.J., DeLuisi, J.J., Mateer, C.L. and Wuebbles, D.J., 1990. Effects of autocorrelation and temporal sampling schemes on estimates of trend and spatial correlation. *Journal of Geophysical Research: Atmospheres*, *95*(D12), pp.20507-20517.

in the section.

l.275: The distribution with s and k should be shown in Fig. 4. We show normal distributions (s=0, k-3=0 by definition) to highlight the non-Gaussian behavior in the ozone residuals, since the assumption that residuals are normally distributed is required in other methods to calculate the time of emergence, unlike the one we use here.

l.345: AK-adjusted is better than "retrieval-adjusted" We use 'kernel-adjusted' to avoid defining additional acronyms.

l.356: errors cannot be negative, but trends can be. The standard deviation is not unexpected to peak near the ozone peak. In this context, the error is the relative difference between two variables, which is at times negative. We now make a mention of this when discussing negative errors, to avoid confusion. Regarding standard deviation: relative errors on the time of emergence are independent of it, see a detailed discussion in the paper which describes the ToE method we use: https://doi.org/10.1029/2024GL109638.

l.381: "... this analysis shows why trends should be analyzed as vertical profiles rather than at individual vertical levels." I do not understand what is meant here. Do you mean that trends should not be evaluated at a single altitude level, but for all levels? Does this make sense? Please

clarify. RC3 had a similar comment. Our statement was meant to echo the result that SBUV kernels can distort the trend profile; we've replaced it with a more general comment about the importance of accounting for averaging kernels when analyzing trends.

l.410: Of the Limb datasets only two use MLS (GOZCARDS, SWOOSH), rather say hat the LOTUS mean trends are heavily weighted by trends from the higher vertically resolved merged limb datasets. Done

Figure 9: Probably legend is wrong (no shading for "undetectable"). It seems that the "undectable" and "undectable according to SBUV" are very similar and differ only by a few tenths of a percent/dec for most altitudes. I think this should be mentioned. Thank you for spotting this error; we modified the figure to ensure that legend and shading match. We have also indicated in the description of Figure 9 that the absolute changes in the detectability envelopes are small.

l.435: Regarding Antarctic column ozone recovery, add some references here. We added a reference to Chapter 4 of the WMO 2022 Ozone Assessment Report.

l. 444: emphasize in item 1, that this is only true for nadir-derived ozone profiles. Done

l. 457: "large in recent years". Are you referring to Hunga-Tonga. Please specify the large events. Following a similar comment by RC1, we changed this to 'which can be large'.

l. 462: add some references dealing with size and timing of the ozone hole. We added a reference to Chapter 4, Section 4.4.2.1 of the WMO 2022 Ozone Assessment Report.

Fig. 4: Both pre-industrial and 500-year runs are labeled ESM4 in the panels. Use different abbreviations for each run. The pre-industrial run is 500 years long (see Section 2.1). We added 'pre-industrial' again after '500 years' in the legend to avoid any confusion.

Figure 5a: More common unit for profiles is DU/km (equivalent to number density). Clarify here. Done.

Fig. 6: add the corresponding SBUV ozone profile to the right panel. We are unsure which panel you are referring to, and have therefore made no changes in response to this comment.

Fig. 7. What are the different conditions between the three panels? different zonal bands like in Fig. 8?, but averaging kernels from 42.5 degs only used for the synthetic data? Note that averaging kernels are solar zenith angle (SZA) dependent and SZAs of SBUV measurements are different in the tropics and higher latitudes. We clarified that the three panels show three different, hypothetical trend profiles in the mid to upper stratosphere, as would be seen by SBUV at 42.5N.

---

## Author Comment (AC2)

Response to RC1

Thank you for your thorough comments—please see our response in blue in the text below.

The manuscript uses ozone time series from the SBUV satellites along with model simulations from the Earth System Chemistry Climate Model (ESM) and the Chemistry Climate Model Intercomparison (CCMI) to investigate long-term trends in ozone and compare them with ozone variability from the model simulations. There are two major results: 1) wide averaging kernels of observations like SBUV mix information from different vertical levels. This can shift and distort vertical trend profiles. 2) Uncertainty estimates are necessary to determine if a trend is significant compared to natural variability. Again, wide averaging kernels combine information from different altitude levels, and this tends to result in underestimated variability. One example are ozone variations associated with the QBO. These are important for trend estimation in the atmosphere, but are reduced and smeared out in SBUV data. This tends to result in errors when accounting for the QBO, and in incorrect uncertainty estimates. Overall this is important information. The paper is well written and deserves publication in ACP.

There are a number of points that should be improved, though:

Abstract and other places in the text: The authors point out a number of problems with merged satellite records (sampling, calibration, instrumental differences, ..). I find this misleading, because the manuscript does not account for any of these "merging" issues. The only issue addressed here, are the SBUV averaging kernels. So I don't think the merging issues should be mentioned in the abstract. The first two sentences should be dropped. In line 4, "one merged" should be replaced by "the SBUV MOD". In line 11 "merged satellite records" should be replaced by "records from instruments with wide averaging kernels". We understand the potential for a misunderstanding, and have modified the abstract to better reflect our focus (in particular, we changed "one merged record" to "one record"). We also mentioned "broad nadir averaging kernels" in that sentence, implemented the requested change at line 11, and changed the title to explicitly mention nadir viewing geometry. We do not mention SBUV MOD in the abstract as we prefer to avoid defining acronyms there.

Line 16/17: "continued recovery" I think this should be "beginning recovery". This also applies to other places where "recovery" is mentioned. We are just at the beginning of ozone recovery. We are far from "recovered" and, as explained in the paper, we are also far from significant recovery in many regions of the atmosphere.
We enacted this change.

Lines 20,22: delete "lower" and add "in the tropics" after "abundances". The main branch of the Brewer Dobson circulation transports ozone rich air in the mid- and upper stratosphere from the tropics to the extra-tropics. Enhanced upwelling in the tropics is decreasing ozone in the tropical lower stratosphere.
Implemented as suggested.

Line 29: suggest to replace "newly detected" by "recent illegal" We changed this to "recent noncompliant," as in the WMO report, rather than "recent illegal."

Line 30: "increasingly large" is too strong. I would say "possibly increasing" Done

Line 32: explicitly add "e.g. Hunga Tonga-Hunga Ha'apai in 2022". Done

Lines 46-47: Reword. You are not adressing merging challenges, you are only adressing the effects of wide averaging kernels. Done: "we turn to the nontrivial effects of error propagation in algorithms used to retrieve ozone abundances from space-borne nadir measurements."

Figure 1: Please explain why the power density of the ESM4 historical runs in the 2 to 20 year range is lower in the top panel and larger in the bottom panel. I guess it is due to applying the SBUV averaging kernel in the top panel. I think this needs to be said / clarified in caption and text. I looks like the averaging kernels reduce variability. Also change the text in the legend in the top panel e.g. to ESM4 historical@SBUV resolution. It needs to be different from ESM4 historical in the legend of the lower panel. The difference in the ESM4 historical run PSDs between the top and bottom panels is indeed due to differential sampling. In the top panel, the historical run is sampled with a 13-year sliding window, while a 165-year window is used in the bottom panel. As a result, the spectral resolution is much coarser in the top panel and several frequency peaks are not resolved in the 2-20 year range, as you point out. We note that no SBUV kernels were applied at this stage; only the temporal sampling is considered. We clarified this in the legend.

Line 98: better to say "pre-industrial simulations" instead of "these simulations" Done

Line 134: I would start a new paragraph after NOAA. It should also be pointed out here that SBUV-MOD and SBUV from NOAA have wide averaging kernels and use the same nadir-viewing satellite data. On the other hand, GOZCARDS, SWOOSH, and the other data sets use LIMB and occultation instruments, which have much finer altitude resolution. We agree on the importance of the distinction between the two categories; to avoid short paragraphs or mentioning related datasets across two paragraphs, we use a bullet list instead.

Figure 2: It would be good to have another panel showing the a-priori ozone profil and the two profiles, in addition to the panel shwing the deviation of the two profiles from the a-priori. The differences between a priori and raw/kernelized model ozone profiles are relatively small (especially since in this case, the a priori is equal to the mean), and therefore difficult to visualize as is (see attached plot). This is why we show the profiles as departures from the a priori, so as to best reveal the effect of the

[Figure]

averaging kernels (see caption). To address your point, we've added the order of magnitude of the errors shown in the caption of Figure 2.

Line 230: for clarification, after "70 hpa", add "from the model simulations"? As shown later the model QBO is quite different from the "real" QBO. We adopted this suggestion.

Line 233/234: "see NOAA ... June 2024" Again, I am assuming you are using the ENSO of the model simulations, not the "real" ENSO. So, while the NOAA page is a good reference, it is kind of misleading here. Please move, reword / clarify. The NOAA page also details how to calculate the ENSO index, but we agree it could be confusing so we removed it.

Line 275: It would be helpful to explain skewness and kurtosis a bit more here. What you are saying is that the residuals are often not normally distributed, with distributions leaning to the left (skewness greater than 0.5), and distributions that are narrower than a normal distribution (positive excess kurtosis). We rephrased to clarify. Note that O3 residuals can have negative or positive skewness, so we explain that the residual distributions are often asymmetric.

Line 294: would be helpful to add "(e.g. the red curve in Fig. 4c)" after "earlier", and " (the black curve in Fig. 4c)" after "itself". Done.

Figure 5: I find it difficult to see much in panel a.) I think it would be better to show here the ratio (standard deviation)/(average values), i.e. the relative standard deviation, e.g. as percent. The overall ozone distribution (average values) will be well known to the readers. The relative standard deviation (or variability) in percent will be much better to compare, e.g. to trends which are also in percent per decade. If the authors don't want to change panel a.) they should add another panel with the relative standard deviation. We've modified the figure as requested—showing the relative standard deviation is an effective way to visualize internal variability evenly across the globe.

Line 314: add "CCMI" before modeled? I assume you are talking about trends from CCMI here. Correct—done.

Line 353: "sampling and retrieval". The way I understand it from section 2.3 you are using SBUV-MOD monthly zonal mean data. I assume you are also using monthly zonal mean data from the model simulations, but without accounting for the specific times and locations of the individual SBUV measurements. Am I correct? Are you dropping polar night data? My guess would be that your model sampling is the same in both hemispheres / polar caps, so "sampling" differences should not play a role here. You only see differences due to the retrieval / averaging kernels, which mixes and redistributes stuff from different altitudes. But, in my understanding, you do not look at sampling differences, i.e. differences due to the specific times and locations of the SBUV observations. So delete "sampling and". SBUV kernels are not available during polar night (they are NaNs) and we now mention this at the end of Section 3.1.1. Our method therefore *does* capture the annual dependency of the SBUV sampling. However, since we are not using the SBUV orbital data to reproduce the specific sampling times and locations in the model, we changed "sampling" to "measurements." As a note, we are not using the SBUV orbital data because:

1. The pre-industrial model run was not available to us with better than monthly resolution at the time of the analysis;
2. SBUV retrievals are averaged zonally and monthly across a large number of profiles (~1000, see Section 3.1.2), and we assume that accounting for specific times and locations of sampling would have a limited effect on the results (see Section 3.1.2);
3. Applying a SBUV-like sampling in the literal sense to the 500-year-long pre-industrial simulation is difficult because the SBUV record is much shorter. Even using the longest continuous portion of the SBUV record (~13 years) as sampling template could introduce a ~13-year cycle in the 500-year-long synthetic observations. This is also why we use one chosen year of kernels to perform the sampling (see Section 3.1.1).

Section 4.3: What you have done is applied averaging kernels and then done trends (avk -> trend). An interesting question to me is whether doing trends first and then applying the averaging kernels to the trend profile (trend -> avk) would give the same result. For the mean this should be the case, because both averging kernels and trend derivation are linear operations on the underlying data. Not sure what it means for the uncertainties though.
This is an interesting consideration; perhaps kernel operators could be applied to trend profiles in DU/decade (rather than %/decade). However, this would require determining an *a priori* trend profile, that is, a representative trend profile for the record. Determining such a profile is in essence the very goal of the scientific community working with merged satellite records. In addition, it is unclear to us what real world process this would represent—trends derived from O3 retrievals already incorporate the effects of kernels after all. Thus, we have not explored this possibility

Figure 7: not sure what the difference between these three panels is. Are you just assuming three different trend profiles? What is the difference between the left panel and the middle panel? Please explain.
Thank you for pointing this out. The panels differ indeed in the idealized trend profile shown (the location and sharpness of the trend maximum). We clarify this in the legend of the figure and in the text.

Figure 10: Please put a label / title on each of the three panels. Top panel is 1.6 to 1 hPa, middle panel 25 to 16 hPa, bottom panel is total column. Done.

Line 438: replace "the total column" by "some ozone column metrics"? Done.

Line 442: "modelled climate" instead of "climate" Agreed that strictly speaking, the reference is to the modelled climate system, however, we feel that in this more general context it is more appropriate to recall the problem of climate internal variability at large.

Line 443: "large" How large? Give numbers. Overall, the changes in uncertainty / significance don't seem to be very large for SBUV (maybe 0.1 or 0.2 % per decade for trend uncertainty according to Fig. 9, a few years according to Fig. 10). They should be smaller to negligible for the LIMB satellites which have much better altitude resolution. Also in lines 6 to 8 in the abstract, it would be good to give some numbers. Fig 9 is not directly relevant to trend uncertainty; rather, Fig 9 quantifies errors in the smallest detectable trend, as a metric for detectability. Fig 7 does quantify trend uncertainties for hypothetical scenarios, and they can be

large (as large as 1 %/dec, i.e., about 100% errors near local maxima in the 'true' trend profile). Regarding errors on significance, Fig 10 shows that up to 13 years may be needed, and Figs 5 and 6 together show that several decades may be needed in some locations (near 25 hPa in the tropics for instance). We've amended our conclusion section and the abstract to be more quantitative, as suggested. Since our trend error estimates come from hypothetical ozone recovery scenarios, we specify this in the new statement.

Line 450, and also discussion of Fig. 7: You might want to refer to Fig. 3-10 of WMO 2022, which shows the latitude altitude distribution of ozone trends from various satellite records. SBUV-MOD is shown in the top left panel of that Figure. You can very clearly see that the peak of upper stratospheric trends is shifted downwards to about 10 hPa in the SBUV-MOD record, and that SBUV-MOD trends are reduced in the 2 to 3 hPa region. Thank you for the suggestion; we now point out this reference here and in the discussion of Fig. 7.

Line 457: "which has been large in recent years". I would say "which can be large". Compared to Pinatubo in 1991, or El-Chichon in 1982/83, most recent volcanic eruptions, even Hunga Tonga, have only had a small influence on stratospheric ozone. Done.

---

## Author Comment (AC3)

Response to RC3

Thank you for your careful review—please see our response in blue in the text below.

The paper describes analysis using synthetic data from a global model to evaluate the impact that broad averaging kernels have on deriving vertically resolved ozone trends from satellite observations. The topic is important because of society's need to determine the timing of ozone recovery as ODSs decline and to assess the ozone response to greenhouse gases. The paper is focused on one specific aspect of the problem but this aspect is generally relevant because many trend estimates rely on SBUV ozone profile timeseries stitched together from multiple platforms. The results will be of interest to those who compile and/or interpret trend estimates and to everyone who want to know the limitations of published trends.

The paper is clearly motivated and well written. The authors thoroughly describe the caveats to their analysis. I recommend some revisions before final publications, as itemized below.

General comments:

1. The units for the vertically resolved ozone from the model data (DU/layer) make it difficult to compare the simulated ozone with measurements or other models without knowledge of the model vertical grid. Even with knowledge of this, calculations would be needed. Can you show the ozone for these plots in the more conventional units such as ppm or number density? This applies to Figure 4 and Figure 5a. In this study, we are more focused on ozone variability and trends than we are with abundances or data set comparisons. The residuals in Figure 4 are only shown for illustrative purposes, to show the inner workings of the ToE method. In Figure 5, ozone abundances are shown as reference to interpret the other panels. Using vmr or ppm would be equally valid choices, but we initially chose DU/layer because they are the native units of the SBUV data set and the units used when applying the SBUV kernels, and because DU/layer profiles are easy to compare to total column results. However, to address your point, we converted DU/layer to DU/km to remove any effects of the vertical grid.

2. It was not clear what the advantage is of interpolating the model profiles to the SBUV grid before applying the averaging kernels (line 188-189). Doesn't this already remove some of the information about vertical structure that you are trying to identify in your study? The kernels are produced at the resolution of the SBUV data set (see Figure 2 for an example), so the model data *must* be interpolated to that resolution first in order to produce synthetic data. We now explain this in the text, and also specify in the caption of Figure 2 that the model profile shown is interpolated onto the SBUV vertical grid in order to isolate the effects of the averaging kernels.

3. Section 3.2.1 is hard to follow. Variables are defined (b, y, etc.) but the equation is not given. Since the final paragraph of this subsection appears to be key to the results that follow, it is important that it be clear. For example, do you compare the two emergence estimates y or y*? I could not tell which was identified as the time of emergence in Figure 5b-5d. Detailing the formalism of the ToE method here would require a significant amount of space, so we provide an overview + a Figure to illustrate, along with the reference. We reworked this section to include more detail and an example. We also specified in the caption of Figure 5 that the ideal time of emergence shown is y, to match

the title of the color bar, and in Figure 6 that the relative differences shown are calculated as $\frac{y^* - y}{y}$. We hope this will clarify that we compare y to y* (see Section 3.2.2) to quantify errors on the emergence of trends attributable to the averaging kernels. y is the 'omniscient' or 'ideal' ToE, and y* is the ToE corrected for the effects of the SBUV kernels, as stated at the end of Section 3.2.1.

Minor comments:

1. (line 98) By "optimistic" do you mean too low? Correct, we've replaced with 'underestimated'.
2. (line 287) The reminder that one should not over-interpret crossing an arbitrary threshold is appropriate; I'm glad to see it mentioned.
3. (line 320) Maybe I missed it but I think this is the first mention that the time to emergence depends on the magnitude of the trend. This is intuitively known but perhaps should be included in the introduction as one of the factors limiting trend detection. We've added a short sentence to this effect at the beginning of Section 3.2
4. At line 381, you state "Altogether, this analysis shows why trends should be analyzed as vertical profiles rather than at individual vertical levels." This is a good summary of the results shown in Figure 7 but is not quantitative. How do you decide whether that criterion has been met? For example, in Figure 8a the time to emergence is detectable over part of the profile but not all of it. However the text indicates that the trend in the upper levels is identified as detectable. RC2 had a similar comment. Our statement was meant to echo the general result that SBUV kernels can distort the trend profile; we've replaced it with a more general comment about the importance of accounting for averaging kernels when analyzing vertically resolved trends.

---

## Author Response (AR2)

Dear Farahnaz,

Thank you for taking a last look at our manuscript. We have implemented the last reviewer comment regarding the units of Figure 2b (we changed DU to DU/layer as requested). We also implemented your suggestions:

**Public justification (visible to the public if the article is accepted and published)**:
Dear authors,

I am pleased to inform you that your manuscript is accepted for publication in ACP after technical corrections. Please find enclosed the report from one referee. There is only one small technical issue left that should be corrected. Further, I would like to ask you to consider the following technical corrections:

P5, L133: write reference as Bhartia et al., 1993 to avoid double closing parenthesis? Here, I am actually not sure what the Copernicus style is. Maybe you could check their guidelines? Done.
P7, L162-163: Sentence correct? Please check. Sentence is correct.
P7, L180-181: Avoid separation of number and unit. Done across the manuscript.
P14, L318: Why is the text in italic? Please use upright font. Changed to upright font.
P15 and throughout the manuscript: Figure should be abbreviated as "Fig." except if it appears at the begin of sentence. In this case it is written as "Figure". Done, as well as "section" -> "Sect." Following the Copernicus guidelines.
P15, L374-375: Same as for P7, L180-181. We modified this paragraph a bit to shorten and clarify the sentence.
P23, L485: Add also the long version of the instrument names. The full instrument names and their abbreviations are defined in Section 4.4 (L441-449) — is it necessary to add the information here again?
P25, L513: Remove "Disclaimer" (since you obviously have none). Done.
References: Check Copernicus guidelines for references. Journal names should be abbreviated. Bibliography was updated according to Copernicus guidelines (including DOIs).

Best regards, Farahnaz Khosrawi